# An Interpenetrating Polymer Network Hydrogel Based on Cellulose, Applied to Remove Colorant Traces from the Water Medium: Electrostatic Interactions Analysis

**DOI:** 10.3390/polym14235090

**Published:** 2022-11-23

**Authors:** Meriem Mihoub, Salah Hamri, Tewfik Bouchaour, Marcel Popa, Dragos Marius Popa, Lamia Bedjaoui Alachaher, Mihaela Hamcerencu

**Affiliations:** 1Laboratoire de Recherche sur les Macromolécules (LRM), Faculté des Sciences, Université Abou BekrBelkaid, BP119, Tlemcen 13000, Algeria; 2Centre de Recherche Scientifique et Technique en Analyses Physico-Chimiques CRAPC, BP 384, Zone Industrielle Bou-Ismail, PTAPC, Tlemcen 42004, Algeria; 3Faculty of Chemical Engineering and Environmental Protection “Cristofor Simionescu”, Department of Natural and Synthetic Polymers, “Gheorghe Asachi” Technical University, Bd. D. Mangeron, nr. 73, 700050 Iaşi, Romania; 4Academy of Romanian Scientists, Ilfov Str., nr. 3, Sector 5, 050045 București, Romania; 5Pharma Science, 100 Bd de L’Industrie, Candiac, QC J5R 1J1, Canada; 6CQFD Composites, 2 Rue du Maine, 68270 Wittenheim, France

**Keywords:** IPN, cellulose, poly(2-hydroxy ethyl methacrylate), docking method, dye retention

## Abstract

The main objective of this work was the removal of eosin Y and green malachite from an aqueous medium by using a cellulose-based biodegradable interpenetrated network (IPN). The IPN was obtained by the sequenced synthesis method. In the first step, cellulose was crosslinked with epichlorohydrin (ECH). In the second step, the obtained gels were swollen in a reactive mixture solution, which was based on the monomers 2-hydroxyethyl methacrylate (HEMA) and 1,6- hexanediol diacrylate (HDDA). After this, swelling equilibrium was reached through the gels’ exposition to UV radiation. An infrared spectroscopy (FTIR) was used to analyze the bond stretching, which confirmed the IPN’s formation. The swelling kinetics in aqueous mediums with different pH values showed a high swelling at a basic pH value and a low response in neutral and acidic media. The IPNs showed an improvement in water uptake, compared to the networks based on PHEMA or cellulose. The IPN was used to remove dyes from the water. The results showed that a high percentage of green malachite was removed by the IPN in six minutes of contact time. The experimental results were confirmed by the docking/modeling method of the system (IPN/Dye). The different physical interactions between the IPN and the dyes’ molecules were investigated. The interactions of the hydrogen bonds with malachite green were stronger than those with eosin Y, which was in good agreement with the experimental results.

## 1. Introduction

The synthetic dyes and molecules used by the textile industry are difficult to biodegrade under natural conditions, therefore, they seriously menace the eco-system since they contaminate tons of water every day [1,2,3,4]. The presence of these contaminants poses a colossal threat to the lives of humans and aquatic animals [5]. Other wastewater treatment techniques, such as biodegradation, have been reported to be ineffective in the removal of these recalcitrant molecules [6].

Recycling wastewaters is an industrial solution to reduce water pollution and to make it potable [7]. Among the developed methods, the absorption technique is both a physical and chemical process [8,9]. Hydrogels are suitable materials to be used as pollutant adsorbents because they have a high capacity to absorb water and also to perform the selective diffusion of molecules such as dyes, catalysts, enzymes and active substances [10,11,12].

The first interpenetrating polymer network was obtained by Aylsworth and the term IPN was first given by Miller in 1960, in a scientific study about a polystyrene-based network [13]. An IPN is a polymer comprising two or more networks, which are at least partially interlaced at a molecular level, but which are not covalently bonded to each other. The network components cannot be separated unless the chemical bonds are broken [14,15,16].

In the present work, the IPN—which was based on two polymeric networks: cellulose and poly(2-hydroxyethyl methacrylate) (PHEMA)—was obtained through the sequenced synthesis method. Firstly, the cellulose was crosslinked by epichlorohydrin, whereas the 2-hydroxyethyl methacrylate (HEMA) was crosslinked by 1,6 hexanediol diacrylate (HDDA), in the presence of the cellulose network. The chemical structure of HDDA was advantageous, firstly, because it has an aliphatic chain. 

The molecular penetration of HDDA during crosslinking can be favorable, leading to a higher level of crosslinking. Secondly, HDDA gives hydrogels improved properties—such as thermal stability, structural uniformity and low surface areas with low pore volumes—that have an impact on the swelling ratio [17,18,19]. This crosslinker has been used in many scientific research articles [20,21,22,23] and the authors have found that it has a faster reaction kinetic, which leads to homogeneous and transparent networks. Furthermore, the chain length between the two acryl groups improves the flexibility of the network.

Donghee K. et al. [24] studied the influence of the type of crosslinking agents used on the morphology of the polymer particles produced by one-step seeded polymerization. They tested several crosslinking agents, such as ethylene glycol dimethacrylate (EGDMA); allyl methacrylate (AMA); 1,6-hexanediol diacrylate (HDDA); and trimethylolpropane trimethacrylate (TMPTMA). They noted that the highest thermal stabilities of the polymeric networks (evaluated by TGA) were obtained when HDDA was used as a crosslinking agent. The order of the chain length between the two functional groups participating in the crosslinking was AMA < EGDMA < HDDA, and it was observed that the chemistry of the reactive groups and the chain length of the crosslinking agents affected the stability and the morphology of the particles.

Cellulose is a polysaccharide that comes from many abundant and renewable resources, making it ecological and sustainable [25]. This polysaccharide is a hydrophilic biopolymer, with a contact angle in the range of 20–30 degrees and its hydrogels can be easily prepared. However, it has a poor mechanical resistance, especially in very low or very high pH media. In order to improve it, PHEMA, a synthetic polymer with good mechanical properties, has been put forward and combined with cellulose to form an IPN [26,27].

Several studies on IPNs based on cellulose and polyacrylate/polymethacrylate [28,29,30] have shown that these materials have high toughness, high strength and good stability. These properties are not observed in the individual polymeric networks, because these ones are held together by physical interactions and permanent entanglements, such as hydrogen bonds.

Malachite green (MG), N-methylated diaminotriphenylmethane, is one of the most common dyes used in the textile industry. It has also been used as an effective anti-bacterial, anti-parasitical and anti-fungal agent. However, it can cause severe effects on the nervous system, liver, brain and kidneys [31]. Due to its negative environmental impact and its carcinogenic effects on mammalian cells, there is a significant interest in developing a new material to remove this dye from water.

Eosin Y is one of the xanthenes dyes. It is a heterocyclic compound containing bromine atoms and a single carboxyl group. It is a water-soluble dye and is presented as a red crystalline powder and it is frequently used in textile dyeing and ink manufacturing. Eosin Y is widely used in biological research and in the pharmaceutical and textile industries because of its strong dyeing properties and its bright color. Eosin Y can biologically degrade once in contact with the environment and can, thus, become toxic to insects. Because of its good water solubility, it also has a significant and negative impact on aquatic life [32].

These two dyes are still largely used in the textile industry, meaning that the wastewater after industrial production is a mixture of dyes and water. Unfortunately, the elimination of the wastewater often occurs through discharge into rivers and other effluents. This was our first motivation for treating the water contaminated by these dyes.

Therefore, the obtained IPN was used to remove the malachite green (the cationic dye) and eosin Y(the anionic dye) from the water.

Many research papers have reported using of different types of hydrogels for dye removal from water [33]. The interaction between dyes and polymers is considered as a critical aspect of this because the attraction between the atoms of the polymers and dyes is responsible for the amount of dye that is retained. In this work, a simulation method based on the molecular docking technique was applied to find the interaction types and to understand the interactions in the polymer/dye system. This simulation method is largely used in the field of biology to explain the interaction between ligands and enzymes [34]. In a previous research article [35], the crosslinked PHEMA-co-HDDA was used for the removal of eosin Y and green malachite pollutants. According to the obtained results, this network showed a low absorption capacity for eosin Y and a high absorption capacity for green malachite. The docking method was applied to analyze these interactions between IPN and dye in the system.

In this work, IPNs based on cellulose and PHEMA were synthetized. The chosen and used pollutants were malachite green and eosin Y. 

Cellulose was chosen for its quality of being a biodegradable material and, thus, it complies with the current environmental laws. In addition, it has a high water swelling degree, which allows a better diffusion of dye molecules through the polymeric network. The first results showed a remarkably high absorption of malachite green but a low absorption of eosin Y. 

Avogadro software was used to draw the three-dimensional (3D) molecular structure. The energy of the system was minimized to obtain a stable state and then the IPN/dye system was treated using AutoDock Vina to find a stable conformation of the system. UCSF Chimera software was used to measure the distances between the macromolecules of the IPN and dyes (oxygen, hydrogen, bromine and nitrogen). These distances allowed us to determine the interaction’s nature and type, namely whether it was strong, medium and/or weak.

This study allowed us to compare the retention of anionic and cationic dyes, which made it possible to establish which dye presents the best interaction with IPNs and, therefore, to establish which IPN is a better absorbent [36]. The presence of cellulose in the IPN enhanced the absorption of malachite green cations. The dye solution becames colorless and the model simulation showed a strong hydrogen-bond type interaction. Compared to malachite green, eosin Y was poorly adsorbed by the IPN. This simulation showed the weak interactions between the IPN and the eosin Y anions, which is explained by the longer inter-atomic distances.

## 2. Materials and Methods

### 2.1. Materials

α-Cellulose (powder); epichlorohydrin (ECH); eosin Y, malachite green; and the 2-hydroxyethyl methacrylate (HEMA) monomer, exhibiting a purity of 97% (all from Sigma-Aldrich), were used as received, without any purification. The 1,6- hexanediol diacrylate (HDDA) monomer, with a purity of 98%, was supplied by Cray Valley (France) and 2-hydroxy-2-methyl-16-phenylpropane-1 (Darocur 1173) was received from Ciba-Geigy. 

### 2.2. Synthesis of IPN

The IPN based on cellulose/PHEMA was obtained through the sequenced synthesis method, in two steps, according to the experimental protocol shown in Figure 1.

First, a cellulose solution (200 g) was prepared from 5% (*w*/*w*) of cellulose, 18% (*w*/*w*) NaOH (12M) [37] and 77% (*w*/*w*) distilled water.

The cellulose powder was suspended in 77 mL of water (quantity necessary to ensure the desired concentration of the suspension), which was left to swell for 2 h, at 5 °C. A separate aqueous solution of NaOH was obtained at a concentration of 18% (*w*/*w*). The two solutions were then mixed using magnetic stirring, for 1 h, until a homogeneous solution was obtained.

Figure 2 shows the first stage of obtaining the IPN, respectively the scheme of the cellulose crosslinking reaction with ECH in the presence of NaOH.

Four different amounts of epichlorohydrin (ECH) were added to cellulose solution of the same weight (50 g), to obtain different weight ratios of the crosslinking agent to cellulose (1%, 7%, 10% and 15%). The obtained solutions were then stirred magnetically for 48 h to obtain the gel. The gel was washed several times with distilled water to remove all the NaOH traces and the excess ECH, until pH values of washing waters were constant. After filtration, hydrogels were put into cylindrical Teflon molds and heated for 24 h, at a temperature of 80 °C, to remove all the water [38]. After drying, opaque hydrogels in disk shape, with a diameter of 18 mm and a thickness of 0.2 mm, were obtained.

Then, the hydrogels (approximately 0.2 g) were immersed in a beaker containing 2 g of crosslinking solution, composed of weight fractions of 99.4% HEMA monomer, 0.1% HDDA as crosslinker and 0.5% Darocur as photoinitiator, at room temperature. The hydrogels were left in the solution until the swollen equilibrium was reached. Then, they were removed, wiped on their surface with filter paper and, finally, were put under UV radiation for 1 h. Thus, the HEMA monomers were crosslinked by photo-polymerization, in the presence of cellulose network.

Figure 3 schematically represents the photochemically initiated crosslinking reaction of HEMA by radical copolymerization with HDDA.

Four IPNs were prepared (noted as a, b, c and d) with different ratios (of epichlorohydrin and HEMA), as they are presented in Table 1.

FTIR spectroscopy was used to establish the necessary duration of photo-crosslinking. At different reaction times, the decrease in the absorbance of the specific band (ν = 815 cm^−1^) of the double bond –C = C- (which was due to the presence of the acrylate groups of the cross-linker and HEMA) was measured (Figure 4). The FTIR kinetic was obtained by analyzing the samples of the cellulose hydrogels swollen in HEMA solution, while it was photo-polymerized at the following time points: before polymerization, after 30 min, after 1 h, and stopped when the C = C bond almost disappeared. 

We observed that after half an hour the absorption decreased and after 1 h it nearly disappeared, confirming the consumption of unsaturated bonds by the polymerization, and, therefore, the IPN’s formation [39]. Consequently, all four IPN’s were obtained at the established duration of photo-polymerization of 1 h. 

### 2.3. FTIR Spectroscopy Characterization

The FTIR spectra of solid samples placed on NaCl plate were recorded within the 4000–400 cm^−1^ range, with the resolution of 4 cm^−1^, and the number of scans was set to 16. All spectra were recorded at 20 °C. The samples were analyzed at initial time by infrared spectroscopy (Perkin Elmer 2000 FTIR model, transmission mode) [40].

### 2.4. Swelling Measurement

Pre-weighed, dried IPN samples were immersed in aqueous media with different pH values, or in HEMA reagent solution (HEMA, 99.4%; HDDA- 0.1%; and Darocour −0.5%), at 20 °C. The ratio between the volume of aqueous media (HEMA reagent solution) and the weight of sample was 10/1 (*v*/*w*): hydrogel samples were 0.40 g to 0.80 in weight. At precise time intervals, IPNs were weighed after removing excess of solvent from the sample’s surface with super-absorbent tissue paper, until there was no change in IPN weight [41,42]. By the swelling of cellulose hydrogels in HEMA reagent solution and the swelling of IPNs, PHEMA and cellulose hydrogels in aqueous medium of different pH values were determined.

The swelling ratio was calculated using Equation (1), as follows:(1)$$Swelling\;ratio\;\left(\%\right)=\left(\frac{W_{s}-W_{d}}{W_{d}}\right)\times 100$$
where, *W_s_* and *W_d_* are the weight of swollen and dried polymer, respectively. Swelling measurements were repeated three times to confirm the results [43].

### 2.5. Dye Retention Measurement

To examine the retention of dyes by the IPN, colored aqueous solutions of malachite green and eosin Y were prepared, in which the IPNs were immersed and the absorption kinetic was studied using UV-Visible spectrophotometer. The IPN hydrogel weight was 0.5 g. The concentrations of malachite green dye solution and eosin Y dye solution were 0.005 mg/mL and 0.003 mg/mL, respectively. The absorption was studied in neutral pH solutions. The spectra were obtained using a double beam Varian Cary (Cary 100) spectrophotometer. Approximately 3 mL solutions were taken at different time points and added to a quartz bowl, and the absorbance values were recorded.

### 2.6. Scanning Electron Microscopy

IPNs were characterized by scanning electron microscopy (SEM) to determine their surface morphology. They were dried, metalized with gold using a spray deposition device, and analyzed using a HITACHI SU 1510 electron microscope (Hitachi SU-1510, Hitachi Company, Chiyoda City, Tokyo, Japan).

## 3. Molecular Modeling of the System—Theoretical Section

### 3.1. Model Description

To model the IPN synthesis process, the HEMA was represented by one molecule; however, the cellulose network was made of two chains—each one contained six monomers of anhydroglucose (AGU) and the chains were bonded by the epichlorohydrin (ECH) molecules.

At a low degree of cellulose crosslinking, with 1% of ECH, two epichlorohydrin molecules were represented. At 15% of ECH, five ECH molecules between the two cellulose chains were represented. The number of ECH molecules were hypothetically proposed as follows: 02 ECH molecules for a non-dense network and 05 ECH molecules for a dense network.

To simulate the dye retention, the models of the system’s IPN/dye molecule were considered. The IPN model contained two networks: the PHEMA and the cellulose network.

The PHEMA network was made of two chains, each composed of seven HEMA repeated units for each and of three HDDA molecules as crosslinkers. It was assumed that the number of chains and molecules represented an approach of the studied IPNs, however, the purpose of the modeling was not for a quantitative analysis. The goal was to see how chemical functions, such as -OH, react in the system (hydrogel/dye) by forming hydrogen or other bonds—it was an atomic simulation.

The cellulose network was described in the elaboration model, as it was an original model that was created in our laboratory.

The dyes’—malachite green and eosin Y—molecules were represented in three dimensions.

### 3.2. Software

Computer software was used for the following three steps: drawing the molecules, the docking analysis and the distance calculation. These steps are described below.

Avogadro software (version 8.0) was used to draw/design the molecular models of the polymers network; the monomer and the dye molecules; the PHEMA and cellulose hydrogels; the IPN; HEMA; and malachite green and eosin Y. It minimized the energy of each model before investigating the docking modeling. For the geometric optimization, the universal force field (UFF) was used, with steps by update equal to four. The algorithm used was the steepest descent method. The output simulation implied eight conformations and the best conformation of each hydrogel/dye system was illustrated based on their energy. The Auto Dock Vina software version 1.5.6 was used to analyze the IPN/dye complexes and presented many conformations of the dye. The dye molecule was considered as flexible while the receptor was kept rigid during docking. Almost all the docking programs have adopted this methodology, such as Auto Dock—which incorporates Monte Carlo-simulated annealing—and evolutionary, genetic and Lamarckian genetic algorithm methods—which model the ligand flexibility, while keeping the receptor rigid [44].

UCSF Chimera software (version 1.5.3) was used to measure the inter-atomic distances between the atoms of the system.

## 4. Results and Discussion

It is well known that the majority of polysaccharides are characterized in their dry state by the rigidity of their chains; therefore, their networks, which are made by covalent bonds, must be even more rigid. In general, polyacrylates are rigid polymers, but HEMA is an exception due to the larger size of its substituent (-O-CO-CH2-CH2-OH), which slightly increases the flexibility of its chains; therefore, PHEMA is used for applications in ophthalmology, e.g., in contact lens production. On the other hand, the covalent networks based on PHEMA are rigid structures, with important mechanical properties.

Cellulose hydrogels are quite fragile and easily fractured upon compression, according to the literature. However, the crosslinked PHEMA is not fragile but can be broken by applying pressure. Numerous articles on cellulose- and acrylate-based IPNs [45,46,47,48] report that the mechanical properties of cellulose could be significantly improved by introducing the acrylic polymer. The same effect was observed for the obtained IPNs in this study, as the IPN became very rigid and could not be broken easily. The more PHEMA the IPN contains, the more hard and rigid it becomes, and it has a low swelling equilibrium ratio, probably because the density of its network makes the diffusion of water molecules difficult.

Another observation made in this study was that the single cellulose and PHEMA networks shrink at the same high pH, while the IPN (cellulose/PHEMA) keeps its original shape intact in alkaline pH, thus, indirectly confirming that IPNs have a better mechanical resistance, compared to single polymer networks.

### 4.1. Structural Characterization of IPNs

The obtained IPNs were characterized from a structural point of view by FTIR spectroscopy (Figure 5).

An absorption band at 668 cm^−1^ specific to -OH groups and a wide absorption band in the range of 3200–3400 cm^−1^ (also attributable to hydroxyl groups) were observed. Furthermore, the absorption band at 1021 cm^−1^ (in the spectrum of the hydrogel that was based only on the polysaccharide) was slightly displaced in the spectrum of the cellulose–PHEMA network. The evidence of these absorption bands in the spectra proved the presence of cellulose in the IPN.

On the other hand, in the spectrum of the hydrogel based on PHEMA, as well as in the spectrum of the IPN, the absorption band attributed to the –C = O bond (1691 cm^−1^) was observed. This bond is explained by the presence of the ester moiety of HEMA and the HDDA. Additionally, those attributable to the –CO- bond (1142, slightly displaced towards 1158 in the IPN spectrum), as well as the peak located at 1063 cm^−1^, proved once again that the natural polymer (cellulose), as well as the synthetic polymer (PHEMA), were found in the composition of the IPN. Based on this, a schematic structure of the network was proposed, as shown in Figure 6.

### 4.2. Morphology of the IPNs

The morphology of two IPN networks was highlighted by scanning electron microscopy. It can be stated that the morphology of IPNc (Figure 7A) was less compact than that of IPNb, although the polysaccharide network had a higher crosslinking density, caused by the higher amount of EPC used (10% compared to cellulose).

However, the use of a larger amount of HEMA (accompanied by HDDA)—in a proportion of 200%, compared to cellulose—led to a much more crosslinked structure (IPNb) as a whole. This effect was due to the greater amount of the HEMA-based network that interpenetrated the polysaccharide network. Its morphology was more compact, compared to IPNc. This explanation is supported by the value of the degree of swelling at equilibrium—of 273% for the IPNc—which was obviously higher than the value corresponding to IPNb (approx. 115%).

This difference was the result of a higher proportion of cellulose, with higher hydrophilicity, in the composition of the IPN, on the one hand, and, on the other hand, it was obviously due to the higher crosslinking density of IPNb, which correlated with the lower overall hydrophilicity of it as a result of the high amount of synthetic polymer.

### 4.3. Behavior of Networks Based on Cellulose in Different Mediums

#### 4.3.1. Cellulose Network Swelling in the HEMA Monomer

As previously specified, the hydrogel was obtained, initially, based only on cellulose crosslinked with ECH, which was then swollen in the solution containing the components that ensured the crosslinking with the formation of the IPN (monomer, crosslinker and photo-initiator), and then subjected it to photo-crosslinking. Consequently, the ability of this hydrogel to swell in the mixture of monomers, which was used for the crosslinking in the second step, was important because it was necessary to ensure the penetration of the components into the network in a uniform way, thus ensuring the formation of a uniform and homogeneous network. The swelling kinetics for a gel based on cellulose crosslinked with 7% ECH, in a monomer solution (HEMA, HDDA and Darocour), is presented in Figure 8.

As we can see, the swelling equilibrium was practically reached after ten minutes. Even if the network swelled quickly (i.e., it only took 10 min to achieve the equilibrium), their shape remained the same, which indicated its stability in this medium.

The increase in the ECH percentage from 1% to 15% decreased the uptake of the reagent solution of HEMA from 300% to 120%, as shown on Figure 9.

Obviously, this effect was due to the increase in the density of the network as the rate of crosslinking increased, which caused the size of the meshes to decrease and reduced the rate and amount of the reactive solution that penetrated the hydrogel. Therefore, cellulose hydrogels absorb different amounts of the reagent solution containing 99.4% of HEMA monomer, depending on the amount of initial ECH. To explain this result, two models were studied according to the crosslinking degree of cellulose (1% and 15% of ECH), as shown in Figure 9 and Figure 10, respectively.

These interactions between the hydrogen and oxygen atoms of the system were classified according to the distances between the cited atoms, as follows: the inter-atomic distances between 2.5 and 3.1 Å were considered strong interactions, those between 3.1 and 3.55 Å were classified as medium interactions and very weak or absent interactions for distances greater than 3.55 Å [49].

##### Model 1: Cellulose Crosslinked with 1% ECH

In Model 1, shown in Figure 10a, strong interactions were noted (as mentioned in Table 2), first, between (O_21_) of cellulose and (H_10_) of the HEMA monomer and, second, between (O_5_, O_6,_ O_9_) of the HEMA monomer and (H_116,_ H_170_, H_170_) of the cellulose network with inter-atomic distances of 2.08 Å, 2.29 Å and 2.13 Å, respectively, see Figure 10b.

The strong interactions explain the diffusion of the HEMA and HDDA monomers in the cellulose networks with a very high percentage and a high swelling degree (300%), as shown in Table 1 (IPN_a_)

At low a crosslinking degree, the -OH functions of the cellulose had a strong interaction with a large number of the HEMA monomer molecules [50].

##### Model 2: Cellulose Crosslinked with 15% ECH

In Model 2, shown in Figure 11a, strong interactions were noted (as mentioned in Table 3) between O_5_, O_6_ of the HEMA monomer and H_284,_ H_142_ of the cellulose network, with inter-atomic distances of 2.32 Å and 2.98 Å, respectively.

Medium interactions were noted between O_5_, O_9_ of HEMA and H_260_, H_113_ of the cellulose network, with inter-atomic distances of 3.26 Å and 3.34 Å, respectively, and also between O_128_ of the cellulose and H_10_ of the HEMA monomer, with an inter-atomic distance of 3.49 Å. A very weak interaction between O_6_, O_5_ of HEMA and H_263_, H_293_ of the cellulose network, with inter atomic distances of 3.58 Å and 3.70 Å, respectively, was noted, see Figure 11b.

The medium and the weak interactions explained the weaker diffusion and, thus, the lower quantity of HEMA that penetrated the network. As a result, a swelling ratio of 100% of the HEMA monomer, which was a lower percentage compared to the cellulose network (300%).

#### 4.3.2. IPN’s Swelling in Aqueous Media

The behavior of hydrogels in water is very important for the application of these materials as decontaminants because their ability to swell in aqueous media is crucial for retaining the diffusion of ions in the network. The kinetics of the swelling in water of a hydrogel based on cellulose and on PHEMA were studied along with IPN_b_. Figure 12 shows the evolution in time of the degree of swelling of these three hydrogels.

The shape of the curves is typical for hydrogels, characterized by a rapid swelling during the first 200 min, followed by a slower evolution until equilibrium is achieved. As expected, the network based on cellulose swelled more quickly and strongly, given the very hydrophilic character of this polysaccharide. Equilibrium was reached after 1500 min and the maximum degree of swelling reached the value of 558%. These values were in agreement with the literature, where it was reported that the swelling ratio for these types of hydrogels tend to be 450% to 1000% [51].

The lowest degree of swelling was achieved by the network based on PHEMA. In this case, the high rigidity of the segments of the chains between the two nodes of the network of the acrylic polymer, as well as its higher hydrophobicity compared to cellulose, was responsible for this low value (approximately 80%) at equilibrium.

The IPN showed intermediate values for the swelling rate (approximately 115%), which were higher than those characteristic of the network based on PHEMA. These results were in agreement with the literature [51]. On the other hand, PHEMA improved the mechanical properties of the cellulose hydrogel and made it more resistant to high pH values. This was due to the presence of the more hydrophilic polysaccharide, which interpenetrated the PHEMA network. However, the values of the swelling rate were not too high because the crosslinking density of the IPN was higher, as compared to that of the cellulose network and even of the PHEMA network.

Often, the rate of swelling of these networks depends on the pH of the medium. Of course, it also depends on the chemical structure of the polymer(s) that constitute the network.

#### 4.3.3. Swelling Behavior of IPN_b_ in Different pH Values

To study the influence of the pH on swelling, the IPN_b_ sample was chosen. The swelling kinetics of this IPN in different pH media, at room temperature, are presented in Figure 13.

It was observed that in acidic (pH = 2.3) and neutral pH media, the swelling rate was not very high (the maximum value was approximately 80% in acidic pH and 115% in the neutral medium), and reached equilibrium after 500 min. On the other hand, in a strongly basic medium, the rate of swelling increased rapidly with time and reached equilibrium after 72 h. Its equilibrium value was approximately 285%.

If we take into consideration the structure of the IPN, we can see that there are non-ionizable groups, therefore, the hydrogel, as it is, cannot present a pH-sensitive character. However, the results showed, all the same, a different behavior in an acid medium compared to a basic medium. The only plausible explanation for this is that the structure of the IPN contained ester-type groups, either in the HEMA or in the crosslinker (HDDA). In a strongly acidic (pH = 2.3) or strongly basic (pH = 13.8) medium, these groups partially hydrolyze. In an acidic medium, the groups which appear are acidic. Consequently, strong hydrogen bonds are formed between these groups, which makes it more difficult for water to penetrate into the pores of the network and, therefore, the swelling rate is lower (Figure 10).

In a strongly basic medium (pH = 13.8), the carboxylic groups formed by the hydrolysis of the ester groups will neutralize and become –COO^−^ Na^+^ groups and, therefore, will dissociate, forming carboxylate anions (-COO^−^). Electrostatic repulsions between the carboxylate anions of the chains or the segments of neighboring chains in the network will increase the distance between the chains and, thus, the pore size of the network, allowing the penetration of higher quantities of water into the network. This causes more swelling. In conclusion, in a basic medium the IPN will behave like an anionic hydrogel [52]. Moreover, the hydrolysis of ester groups occurs more intensively in a basic medium. Consequently, certain bridges between the chains of PHEMA will break and the size of the meshes of the network will increase, and this explains the higher degree of swelling in the alkaline medium (more than three times higher compared to the neutral or acid media) [53].

Lin et al. [45] presented a study on the swelling of IPNs made of cellulose and poly(acrylamide) hydrogels in different pH media. They reported that the equilibrium swelling ratio of all the IPNs increased gradually with the increase in the pH and reached the maximum value at a pH of 13.0. As a result, the hydrogels displayed an expandable network structure, which could easily absorb water. In an acidic pH medium, most of the carboxylate anions were protonated. Meanwhile, a stronger intermolecular hydrogen bond was formed between the cellulose chains and the PHEMA network, which induced the contraction of the IPN hydrogel and the expulsion of the water molecules [54,55].

### 4.4. Application of IPNs in Dye Removal

Various studies have reported the absorption of cationic dyes by anionic IPNs, containing weakly acidic or strongly charged groups. The combination of electrostatic interactions with hydrophobic interactions is usually the driving force in the case of cationic species [56].

The IPNs based on cellulose/PHEMA were applied to remove malachite green and eosin Y from the water medium. The docking method was used to determine the different interactiins between the IPN and dye in the system. 

#### 4.4.1. Dye Removal by the IPN

The study of the absorption kinetics of the malachite green dye in an aqueous solution showed excellent retention by the IPN. As shown in Figure 14a, the IPN was colored and the aqueous solution was decolorized.

Compared to the malachite green, the study of the absorption kinetics of the anionic dye, eosin Y, showed a weak retention by the IPN (which initially is white), observed by the weak orange coloring of the IPN sample and the fact that the eosin Y aqueous solution remained colored after a long contact time (Figure 15a, right side). As observed, the UV-visible spectra showed a peak at 520 nm, the absorbance slowly decreased in the first hour and then remained practically constant until 140 min (Figure 15b).

The study of the absorption of the dye mixture by the IPN in an aqueous solution showed that a large quantity of malachite green was adsorbed after 6 days, whereas a negligible quantity of the eosin Y was absorbed (Figure 16). This result showed the selective retention of malachite green dye by our IPN, compared to eosin Y.

#### 4.4.2. Explanation of Selective Absorption of Dyes by Using the Docking Method 

The behavior of the IPN/dye system was investigated using this model by focusing on the interactions between the atoms of the IPN network and the dyes’ molecules. 

##### Model 1: Cellulose–PHEMA/Malachite Green

A model of the system’s IPN made of cellulose–PHEMA/green malachite molecules was considered. The distances between the oxygen, hydrogen and nitrogen atoms allowed the determination of the nature and type of the bond.

In this model, shown in Figure 17a, strong interactions were noted, as mentioned in Table 4, between (N_17_) of malachite green and (H_99_, H_637_) of IPN, exactly from the OH of cellulose, with inter-atomic distances of 2.63 Å and 2.67 Å, respectively, and also between (O_629_) of the cellulose network and (H_19_) of the malachite green with an inter-atomic distance of 3.385 Å, see Figure 17b.

The second (N_7_) of malachite green had very weak interactions with the hydrogen atoms of the PHEMA hydrogel in the IPN network.

The interaction was stronger with -OH from the cellulose hydrogel than the interactions with -OH of PHEMA hydrogel in the IPN.

##### Model 2: Cellulose–PHEMA/Eosin Y

The model of the system’s IPN, made of cellulose and PHEMA/eosin Y molecules, was considered as shown in Figure 18a.

Strong and medium interactions have been listed in Table 5. Two strong interactions between (O_9_, O_8_) of eosin Y and (H_99_) of cellulose hydrogel in the IPN network, with inter- atomic distances of 1.89 Å and 2.79 Å, respectively, and a medium interaction between (O_8_) of eosin Y and (H_98_) of cellulose hydrogel in the IPN network, with an inter-atomic distance of 3.53 Å were observed, as shown in Figure 15b.

A weak interaction between (Br_19_, Br_27_) of eosin Y and (H_620_, H_637_) of the HEMA network in the IPN, with inter-atomic distances of 3.89 Å and 3.84 Å, was also measured.

The oxygen of the eosin Y dye molecule was strongly attracted to the hydrogen atoms in group -OH of the cellulose hydrogel in the IPN network, and the Br of eosin Y was weakly attracted to H of the group -OH from the PHEMA hydrogel [57].

The weak discoloration of the eosin Y solution by the cellulose hydrogel was due to many reasons. Firstly, it could have been because the large fraction of the hydroxyl groups were surface-inaccessible due to the inter-/intra-molecular hydrogen bonding between the cellulose and PHEMA networks. Secondly, it could have been attributed to non-specific H-bonding interactions between the saccharides and eosin Y. Finally, it could have been due to the morphological changes in the cellulose materials, which influenced the dye’s accessibility within the micropore domains. In fact there is a semi-quantitative dependence between the level of the ECH content and the level of the discoloration [58,59].

In comparison with previous work, the PHEMA network presented a low retention of malachite green, whereas the present IPN had 100% retention of the dye. For the retention of eosin Y, both the PHEMA network and IPN had low retention; therefore, the IPN used in this work can be applied efficiently to absorb the malachite green dye from the water medium.

## 5. Conclusions

An interpenetrating polymeric network (IPN) was obtained through a sequenced method, from poly(hydroxyethyl methacrylate) (PHEMA) and cellulose networks. Epichlorohydrin and HDDA were used as crosslinkers.

The high swelling of the cellulose network in a HEMA monomer medium has been explained by the docking method. The hydrogen bonding that occurred between the -OH functions of both the cellulose and HEMA molecules increased the diffusion of the HEMA monomer inside the cellulose networks.

The equilibrium swelling obtained by the IPN made from cellulose and PHEMA in the aqueous media, proved the hydrogel character of materials and the fact that the crosslinkers—epichlorohydrin and HDDA—have a good crosslinking effect.

The elaborated IPN was used to remove the chosen dyes—green malachite and eosin Y. A high retention of the green malachite—with a high percentage being absorbed in six minutes of contact time—and a poor retention of the eosin Y were noted. 

The docking method’s outputs showed a strong interaction in the first case and weak interactions in the second case, which explained this behavior. 

The immersion of the IPN in the mixture composed of green malachite and eosin Y showed that the IPN removed the green malachite and left the eosin Y, which underlined the selective removing property of this IPN.

## Figures and Tables

**Figure 1 polymers-14-05090-f001:**
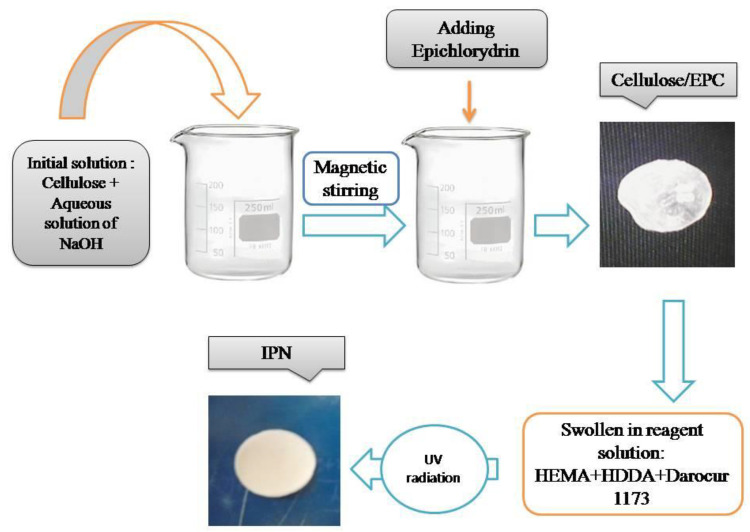
Schematic illustration of the elaboration steps of the cellulose- and PHEMA-based IPN.

**Figure 2 polymers-14-05090-f002:**
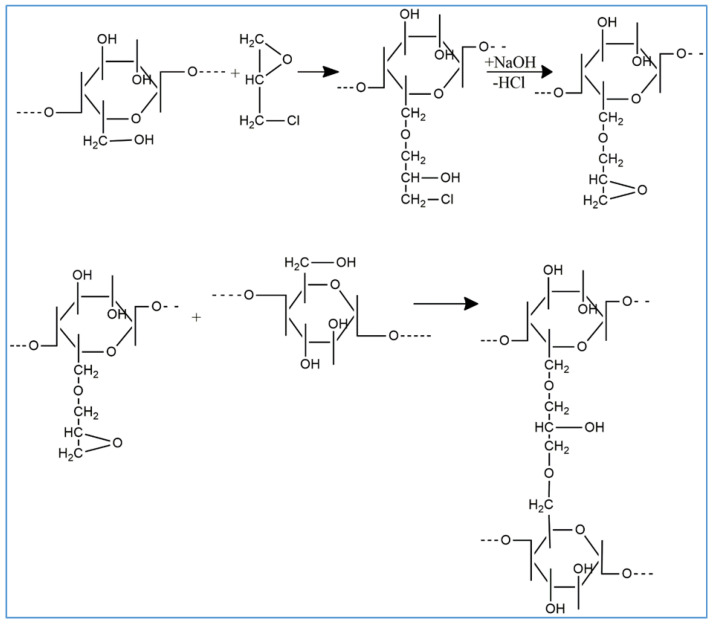
Schematic representation of cellulose crosslinking by ECH.

**Figure 3 polymers-14-05090-f003:**
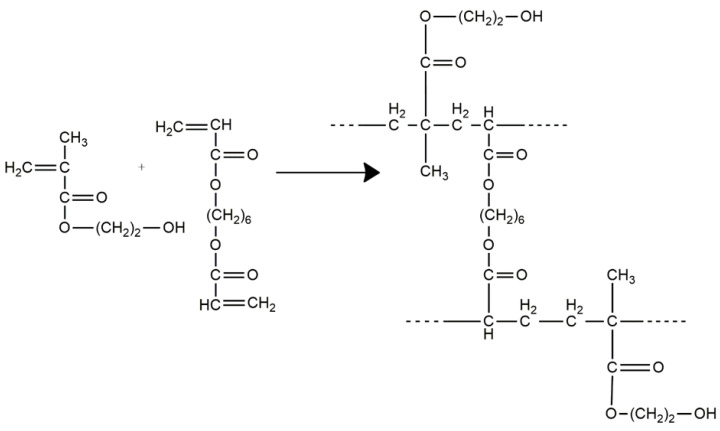
Schematic representation of HEMA crosslinking by HDDA.

**Figure 4 polymers-14-05090-f004:**
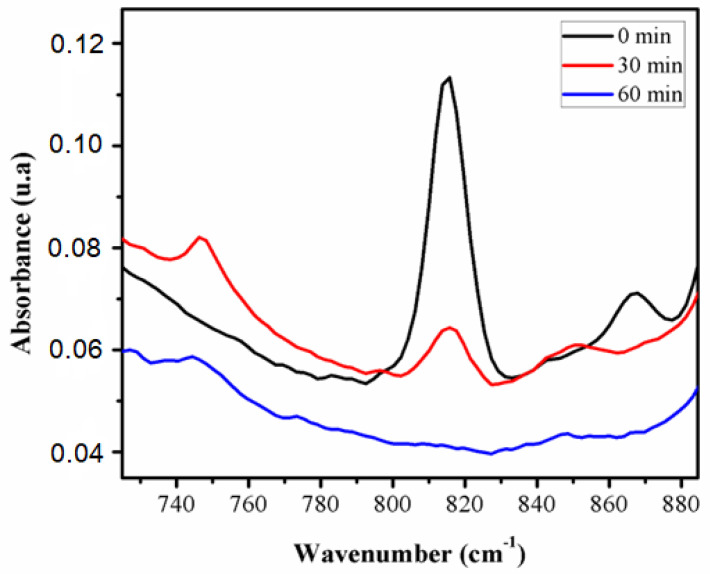
Time evolution of the acrylate double bond absorbance present in the reactive mixture under photo-polymerization.

**Figure 5 polymers-14-05090-f005:**
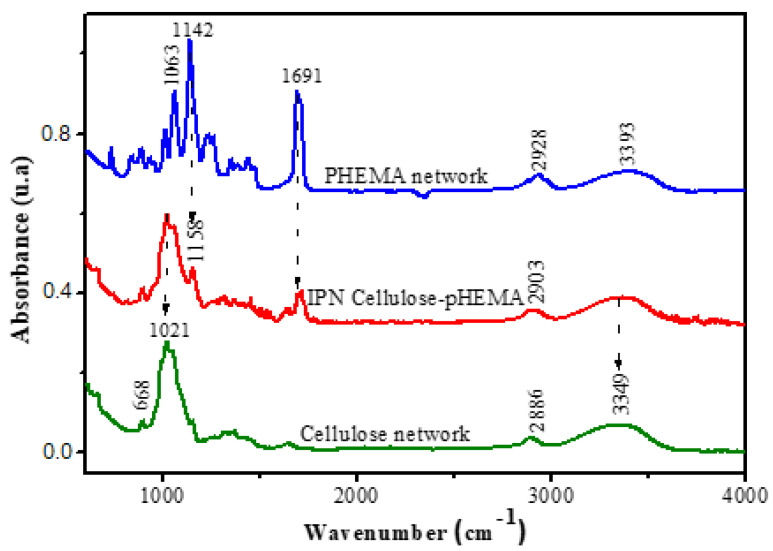
FTIR spectra comparing cellulose hydrogel, PHEMA hydrogel and IPN_b_ (cellulose/PHEMA, 10/90, ECH = 7%).

**Figure 6 polymers-14-05090-f006:**
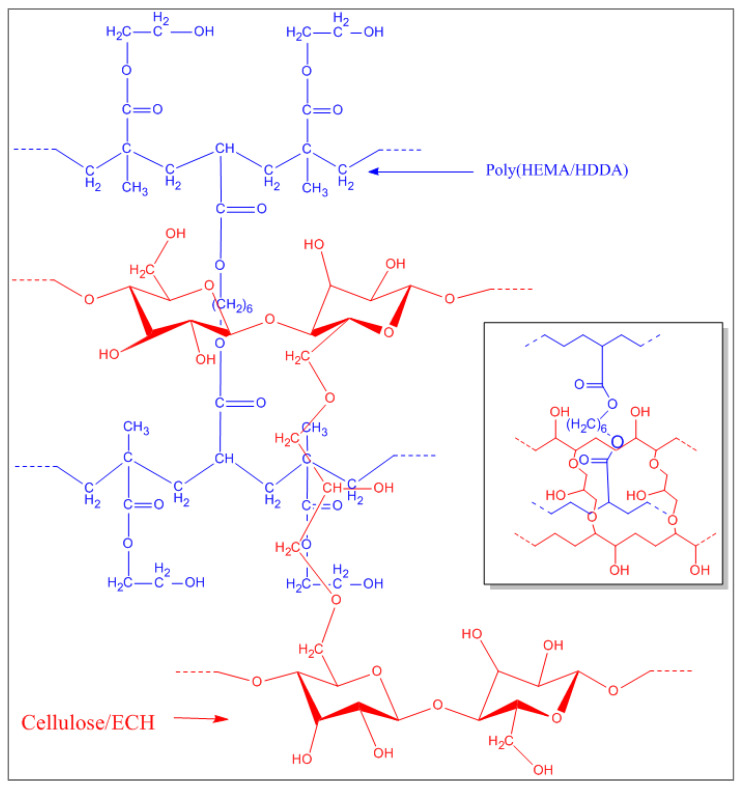
Schematic structure of an IPN based on cellulose and PHEMA, obtained by sequential crosslinking.

**Figure 7 polymers-14-05090-f007:**
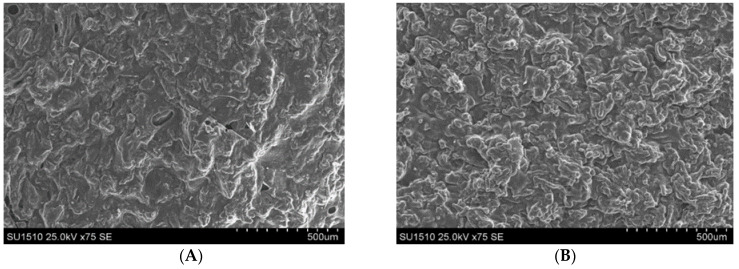
SEM micrographs—comparison of the morphology of IPNc (**A**) and IPNb (**B**).

**Figure 8 polymers-14-05090-f008:**
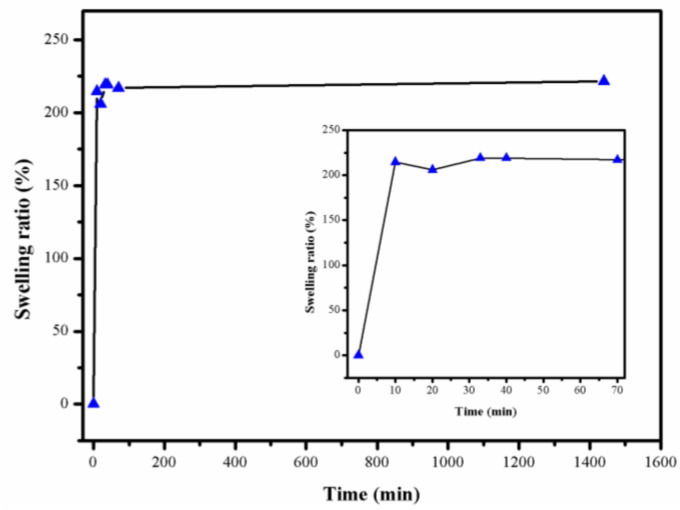
Swelling kinetics of the dried hydrogel cellulose (ECH 7%) in HEMA reagent solution (HEMA, 0.1 HDDA, and Darocur) at T = 20 °C. The inner plot shows aggrandizement for time intervals: 0 to 70 min.

**Figure 9 polymers-14-05090-f009:**
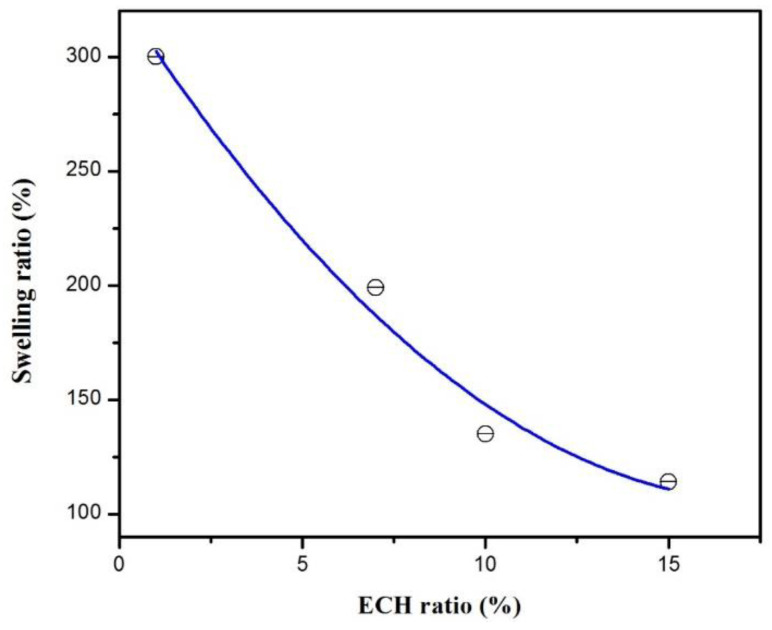
Swelling equilibrium of cellulose hydrogels with different ECH ratio in HEMA reagent solution.

**Figure 10 polymers-14-05090-f010:**
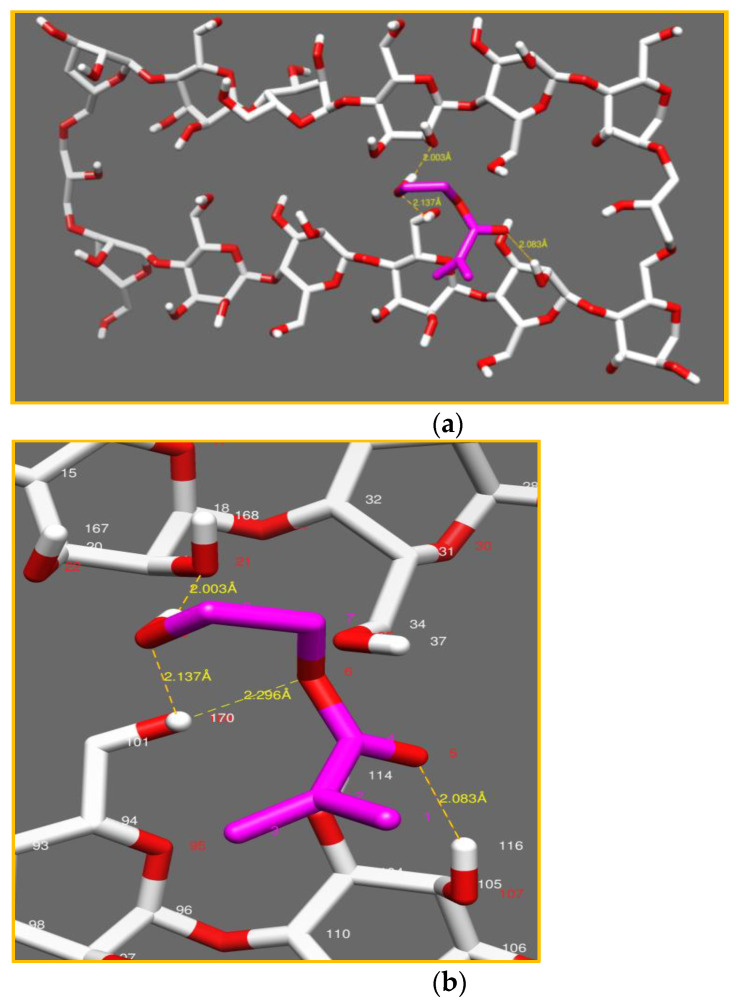
Conformational study of the model (cellulose network/HEMA monomer) with (ECH 1%) (**a**). Aggrandizement of interactions (**b**).

**Figure 11 polymers-14-05090-f011:**
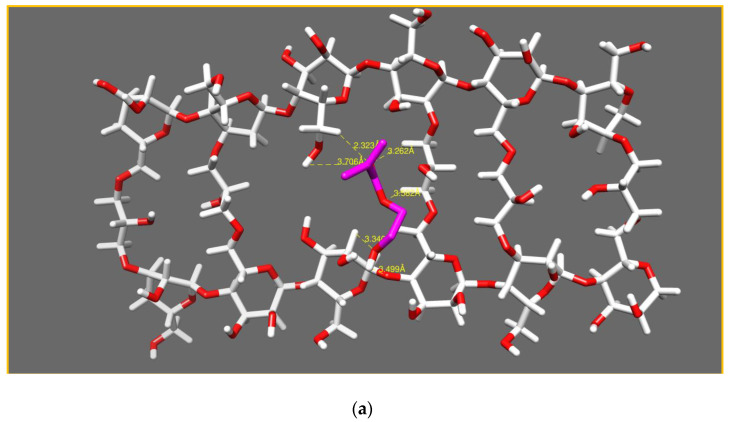
Conformational study of the model (cellulose network/HEMA monomer) with (ECH 15%) (**a**). Aggrandizement of interactions (**b**).

**Figure 12 polymers-14-05090-f012:**
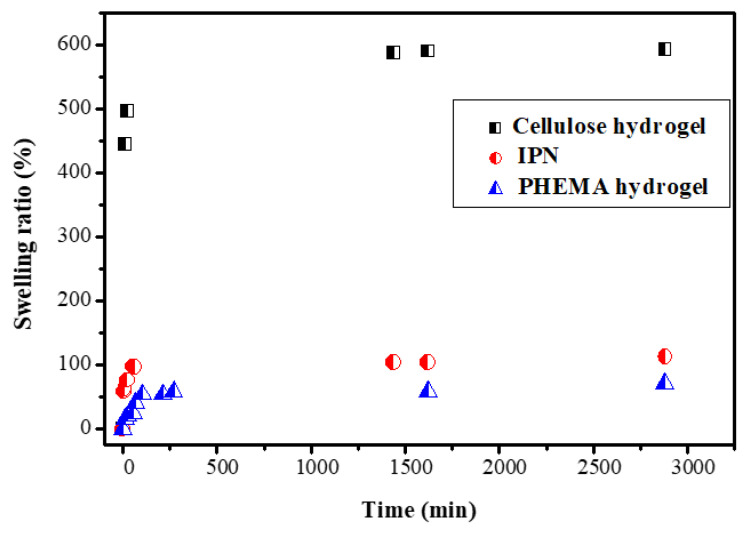
Swelling kinetic in water medium of the IPN_b_ (ECH 7% and HEMA 200%), cellulose (ECH 7%) and PHEMA hydrogels.

**Figure 13 polymers-14-05090-f013:**
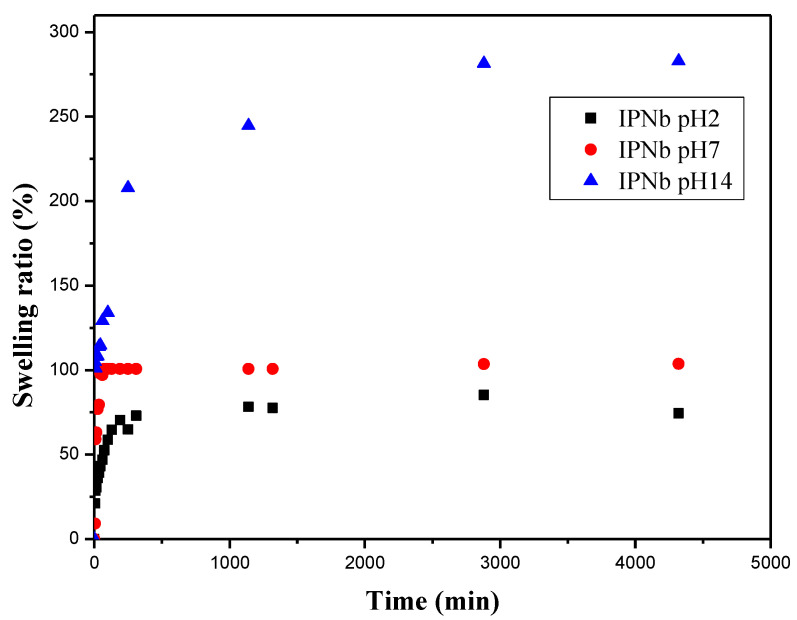
Swelling kinetics of the IPN_b_ with ECH 7% and HEMA 200%, in aqueous media of different values of pH.

**Figure 14 polymers-14-05090-f014:**
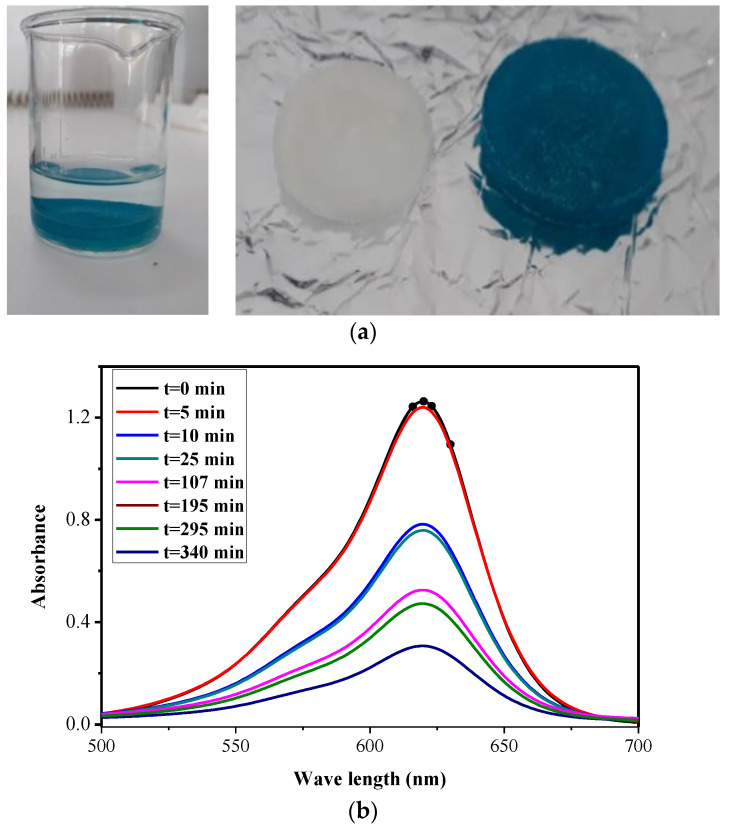
The malachite green solution and IPN hydrogel (before and after dye absorption) (**a**). Absorption kinetic of malachite green by IPN (cellulose–polyHEMA hydrogels) (**b**).

**Figure 15 polymers-14-05090-f015:**
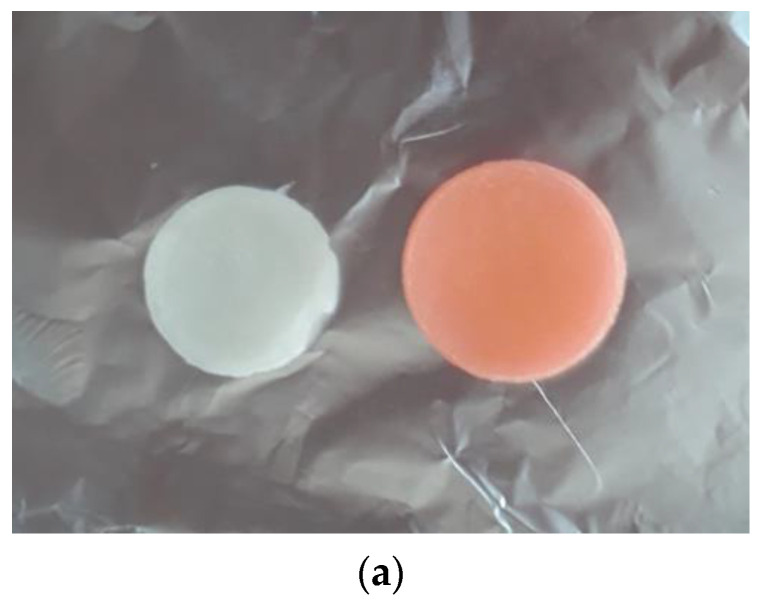
IPN_c_ hydrogel before (left side) and after dye absorption (**a**). Absorption kinetics of eosin Y by IPN (cellulose–PHEMA hydrogels) (**b**).

**Figure 16 polymers-14-05090-f016:**
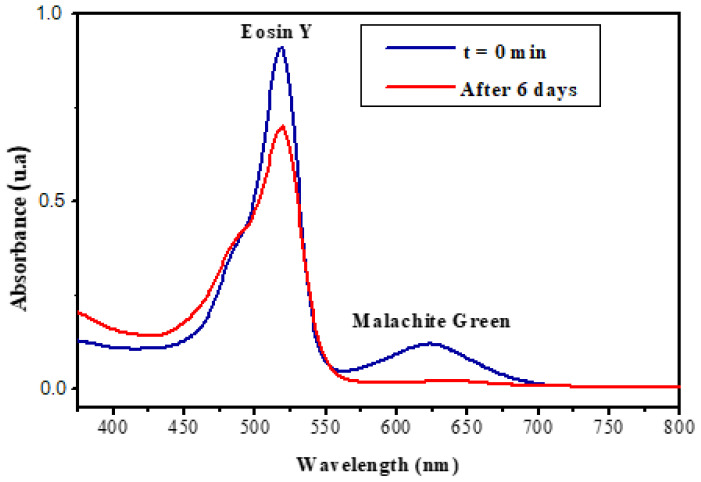
Retention of a mixture of dyes (malachite green at ν = 620 nm and eosin Y, at ν = 520 nm) by IPN (cellulose/polyHEMA) hydrogels.

**Figure 17 polymers-14-05090-f017:**
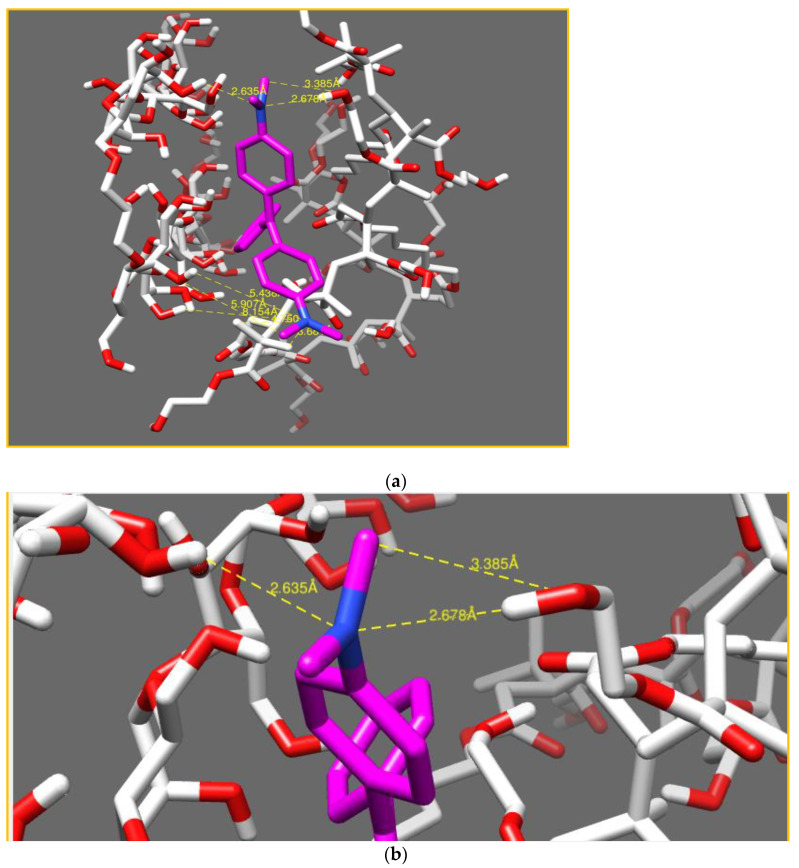
Representation of interactions in the system (cellulose–polyHEMA hydrogels/malachite green-based IPN) (**a**). Aggrandizement of the interactions (**b**).

**Figure 18 polymers-14-05090-f018:**
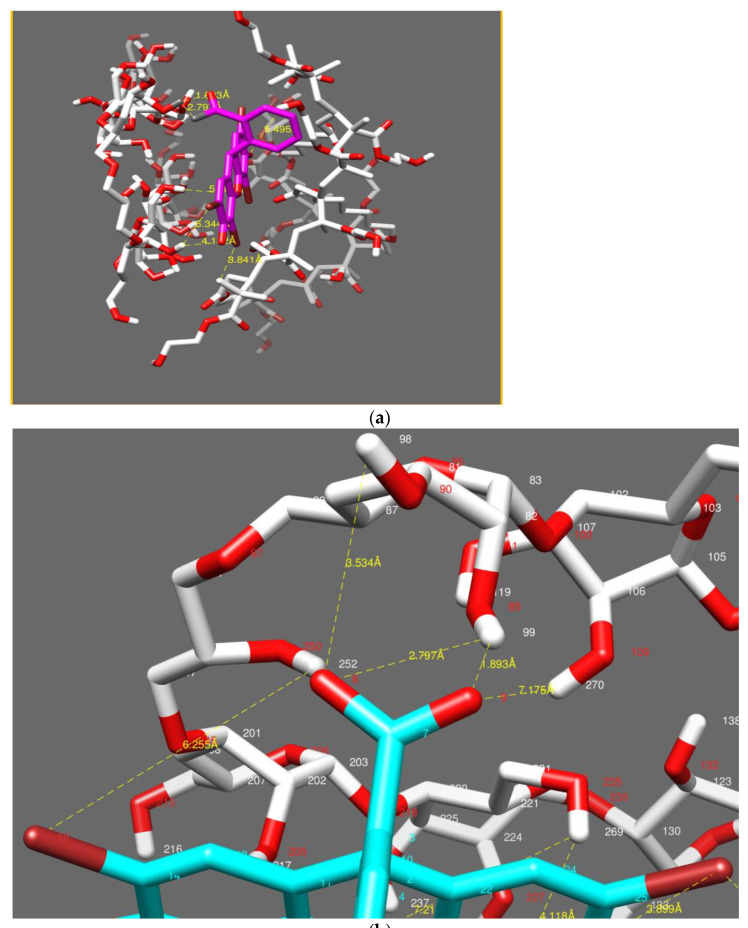
Representation of interactions in the system IPN (cellulose–polyHEMA hydrogels)/eosin Y (**a**). Aggrandizement of interactions (**b**).

**Table 1 polymers-14-05090-t001:** Experimental plan for obtaining IPNs based on cellulose and PHEMA *.

Sample	ECH (w %) (Compared to Cellulose *)	HEMA ** (w %) (Compared to Cellulose Hydrogel)
N_c_	7	0
N_PHEMA_	0	100
IPN_a_	1	300
IPN_b_	7	200
IPN_c_	10	130
IPN_d_	15	100

* The cellulose concentration was fixed for all the samples’ preparations at 5%; ** The HDDA rate was kept constant, with a weight ratio of 0.1, compared to HEMA reactive solution.

**Table 2 polymers-14-05090-t002:** Interactions in the system—cellulose hydrogel/HEMA monomer (ECH = 1%).

Bond	Number	Distance (Å)	Type	Nature
O……H	03	2.00	Hydrogen bond	Strong
		2.13	Hydrogen bond	Strong
		2.08	Hydrogen bond	Strong

**Table 3 polymers-14-05090-t003:** Interactions in the system of cellulose hydrogel/HEMA monomer (ECH = 15%).

Bond	Number	Distance (Å)	Type	Nature
O……H	07	2.32	Hydrogen bond	Strong
		2.98	Hydrogen bond	Strong
		3.26	Hydrogen bond	Medium
		3.34	Hydrogen bond	Medium
		3.49	Hydrogen bond	Medium
		3.58	Hydrogen bond	Weak
		3.70	Hydrogen bond	Weak

**Table 4 polymers-14-05090-t004:** Conformational study in the system IPN (cellulose–polyHEMA network)/malachite green dye.

Bond	Number	Distance (Å)	Type	Nature
N……H	03	2.63	Hydrogen bond	Strong
		2.67	Hydrogen bond	Strong
		3.36	Hydrogen bond	Strong

**Table 5 polymers-14-05090-t005:** Conformational study in the system: IPN (cellulose–polyHEMA network)/eosin Y.

Bond	Number	Distance (Å)	Type	Nature
O^−^ ….H (IPN)	03	1.89	Hydrogen bond	Strong
		2.79	Hydrogen bond	Strong
		3.53	Hydrogen bond	Medium
Br….H	04	2.77	Halogen bond	Strong
		3.84	Halogen bond	Very weak
		3.89	Halogen bond	Very weak
		4.11	Halogen bond	Very weak

## Data Availability

The data presented in this study are available on request from the corresponding author.

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
