# Peer review of "An Interpenetrating Polymer Network Hydrogel Based on Cellulose, Applied to Remove Colorant Traces from the Water Medium: Electrostatic Interactions Analysis"

_polymers, 2022, doi:10.3390/polym14235090_

Round 1

Reviewer 1 Report (Previous Reviewer 1)

The authors have improved the previous manuscript with the new corrections.

Reviewer 2 Report (Previous Reviewer 4)

The presentation quality of this manuscript has significantly improved. I do not have any more concerns on this manuscript now and I would recommend to accept this work.

This manuscript is a resubmission of an earlier submission. The following is a list of the peer review reports and author responses from that submission.

Round 1

Reviewer 1 Report

In this work Mihoub et al. present an experimental and theoretical study on the use of an interpenetrated polymeric network (IPN) to remove some dyes from contaminated waters. The proposed hydrogel based on cellulose was synthetized through different steps of crosslinking and characterized. Also, a molecular modeling study was conducted by minimizing a simple structure made of two crosslinked chains for each polymeric network with an ulterior docking study on the chemical interactions between the IPN and some dye molecules. The authors are presenting interesting study and are very ambitious. Unfortunately, the work is neither well written nor organized, being difficult to be followed by any potential reader. In addition, there are a large number of missing details to facilitate its reproducibility and many statements presented by the authors, which are not backed up by either experimental or theoretical evidence. Similarly, there are external references/observations in the literature that are not cited. 

1. It is not clear the different models built for each modeled system.  Could the authors elaborate more on the way that they build the 3D network model? Also, there is no information about either the force-field (MM optimization) or the QM level (QM optimization) that was conducted in the energy minimization of the system. Which was the minimization protocol used in the systems?

2. what about the docking methodology used?

3. It seems that the system lacks periodicity, and no real 3D network bulk was built as a 3D network model. This is very important to derive realistic results, since the chemical environment using low level docking methodology is very important, which becomes unsound work.

4. The synthetized IPN is under characterized. Additional experiments would be needed, such as Raman spectroscopy, thermal stability and surface morphology (SEM) would be required to define the proposed IPN hydrogel. Only FTIR and swelling studies were conducted for their characterization. In the text the authors are talking about mechanical properties of the new IPN which should be backed up by either a tensile-stress or rheological study.

5. It is not clear the evidence shown by the authors to explain the system behavior under different environmental pHs. Only SR versus pH evidences are shown but no mechanical studies and chemical model interactions were analyzed at different pHs.  

Minor points:

- Pag. 2. There is a paragraph with a different font size.

- Please, check for all the super- sub-indexes along the text.

- Pag. 10, just before Table 2. There are wrong Angstrom symbols, as well as in many sites along the text. Please, check it.

- Table 2 and within the text. Excessive number of decimals for a theoretical distance measurement since they were not obtained through a high level ab-initio calculation (QM).

- The model figures have to be improved since it is difficult to read their included distances as well as their comprehension. 

Reviewer 2 Report

Authors prepared the IPN hydrogel using cellulose and PHEMA for the retention of different dyes from water. They used FTIR anlysis for the characterization of the gel. They also used molecular modeling study to further confirm the experimental results.

However, the characterization of the prepared gel was so preliminary. Further clear analyses for the prepared gel are necessary.

1) Resolution of FTIR is so low, 4 cm-1. More clear analysis of peaks is necessary based on the molecular structures. NMR, FE_SEM, DSC analysis are necessary for the structural characterization.

2) Rheologyl measurements of hydrogel are necessary. Stroage, loss modulus and viscosity measurements are necessary.

3) Compressive and tensile tests are necessary.

4) Equilibrium swelling ratio of the prepared hydrogel should be determined.

Reviewer 3 Report

The paper titled "Interpenetrating polymer network hydrogel based on cellulose applied to remove colorant traces from water medium: electrostatic interactions analysis" presents really interesting results which can be published in Polymers mdpi after major revision. The following issues should be addressed: Please add chemical reactions to illustrate the step-by-step process of the synthesis. Figure 1 and 4 are really not enough to understand how the final structure was synthesized. In the paper are a high amount of typos and grammatical mistakes, all text should be carefully checked. It is completely unclear to me why the authors selected a model of the PHEMA network which was made of two chains each composed of seven HEMA repeated units for each and three HDDA molecules as cross-linker. Please cite relevant works where similar chemical structures were presented. Why malachite green and eosin Y was selected for studies? Appropriate discussion should be provided.  https://doi.org/10.1007/s10965-020-02335-7 https://doi.org/10.1007/s10570-018-2171-y  

Reviewer 4 Report

This is a good manuscript from the scientific perspective, the the content needs to be better presented. In this work, the author synthesized cellulose- PHEMA IPN and demonstrated its application in the selective dye removal. Sufficient characterization were implemented to investigate the produced IPN and simulation was used to explore the interaction of IPN network and the mechanism of selective dye binding. However, the abstract, introduction and conclusion sections need to be reorganized to better delivery the background, rationality/novelty, conclusion and perspective of this work. Besides, extensive attention needs to be paid to the format. 

Taking all together, I would recommend careful revision before pulication.